# High-Performance Work Systems, Thriving at Work, and Job Burnout among Nurses in Chinese Public Hospitals: The Role of Resilience at Work

**DOI:** 10.3390/healthcare10101935

**Published:** 2022-10-02

**Authors:** Zhe Yun, Peng Zhou, Bo Zhang

**Affiliations:** 1School of Economics, Capital University of Economics and Business, Beijing 100070, China; 2Cardiff Business School, Cardiff University, Cardiff CF10 3EU, UK; 3School of Economics and Management, Beijing University of Chemical Technology, Beijing 100029, China

**Keywords:** resilience at work, high-performance work systems, thriving at work, job burnout, nurses

## Abstract

The overall purpose of this study is to explore and examine whether high-performance work systems (HPWS) can impact thriving at work and job burnout via resilience at work among nurses in Chinese public hospitals. Specifically, it draws on social exchange theory to conceptualize a positive relationship between HPWS and resilience at work. Then, based on a socially embedded model of thriving at work and knowledge about job burnout in the literature, it further proposes the differentiated mediation roles of resilience at work in the relationship of HPWS to thriving at work and job burnout. A time-lagged survey involving three rounds of data collection with self-reported online questionnaires was employed. A total of 160 nurses from 20 public hospitals in China were invited to participate in the research. Finally, a sample of 845 responses was obtained. The response rate was 52.8%. Multiple regression analyses were conducted to test the hypotheses. It was found that HPWS (time 1) positively impacted resilience at work (time 2). The results also demonstrated that HPWS (time1) enhanced thriving at work (time 3) and reduced job burnout (time 3) via developing resilience at work (time 2). To conclude, being resilient in the workplace is crucial for nurses to enhance thriving at work, and inhabit job burnout among nurses. HPWS is a valid management tool that can be used to cultivate a nurse’s resilience at work, which further reduces job burnout and enhance thriving at work.

## 1. Introduction

The frontline nurses in healthcare organizations have been experiencing great psychological distress and job burnout, and the case became even worse after the outbreak of the COVID-19 pandemic [1,2]. It is important for both scholars and policymakers to understand how to help nurses to cope with negative psychological states [3]. A strategy to cope with job burnout and thrive at work is known as resilience at work [4], the ability to adapt to, or bounce back from, extremely unfavorable circumstances [5], or the ability to positively adapt to traumatic or adverse experiences [6]. The literature has suggested various ways to cultivate a nurse’s resilience at work. A meta-analytic review found that multiple resilience-building programs had moderate positive effects on resilience [7]. For example, a study reported that healthcare workers’ resilience was strengthened after participating in an evidence-informed resilience-promoting curriculum called R2 [8]. Other factors that promote healthcare workers’ resilience include social support [9], occupational physical activity, moderate-to-vigorous physical activity at work and dynamic standing at work [10], empathy [11], etc. These studies can certainly help us to understand the antecedents of resilience. However, they only focused on a certain specific program’s or management practice’s effectiveness. When managing employees, modern healthcare organizations widely use HRM systems to manage nurses [12,13]. Yet, the impact of the HRM systems on a nurse’s resilience at work is still under-researched. Such dearth hinders us from achieving a better understanding of how to develop a nurse’s resilience at work effectively. Therefore, we explore and examine the influences of HRM systems on a nurse’s resilience at work.

When facing challenges, employees often choose to stay learning and energetic to thrive at work [14] or feel exhaustion, cynicism, and incompetency, namely, job burnout [15]. Job burnout is often reported among nurses in countries such as China, where nurses suffer from work stresses resulting from heavy workload and hospital–patient conflict and hospital management reforms [16]. Empirical studies have reported that burnout among nurses is common in China [17]. To cope with nurses’ stresses and burnout, healthcare organizations are suggested to frequently support nurses in thriving at work in China [18,19,20]. A nurse’s resilience has been reported to reduce negative employee outcomes, such as work stress [21], job insecurity [22], and turnover intention [23], while enhancing positive outcomes, such as proactive work behavior [24], perceived justice, trust in the organization, creative performance [22], job satisfaction, quality of care [23]. Furthermore, HRM systems have also been argued and reported to enhance positive and reduce negative employee outcomes [25,26]. In this case, there might be a mediation role of resilience at work in HRM systems’ relationship to thriving at work and job burnout. Therefore, we explore and examine whether HPWS can enhance a nurse’s thriving at work and reduce job burnout by developing a nurse’s resilience at work.

The high-performance work system (HPWS) is a typical type of HRM systems that is designed to enhance employees’ skills, commitment, and productivity in such a way that employees become a source of sustainable competitive advantage [27]. It has been reported that HPWS is a popular management tool among healthcare organizations [12,13]. This study explores and examines the mediation effects of a nurse’s resilience at work in the relationship between HPWS and a nurse’s thriving at work and job burnout.

## 2. Hypothesis Development

### 2.1. HPWS and Resilience at Work

According to social exchange theory (SET), employees socially interact with coworkers and organizations to obtain social resources and make contributions [28,29,30]. Exchange parties comply with the rules of reciprocity and equivalence. The rule of reciprocity means that the recipient tends to give back a benefit to the entity that offers such a benefit. The principle of equivalence means that the recipient should return benefits of equivalent value [30]. HRM systems, such as HPWS, can adjust the characteristics of employee–organization social exchange relationships to induce employees to develop certain abilities [31,32,33], including resilience at work [34].

The elements of HPWS usually include practices such as extensive selection and recruitment, participative decision-making, extensive training practices, employee ownership programs, results-oriented and team-oriented performance appraisal, profit-sharing programs, etc. [25,26]. When experiencing HPWS, organizational members translate an organizational event and construct a meaningful explanation for that event [35]. For instance, extensive training programs at work can be interpreted as the organization expects employees to enhance abilities in the work-related domain or to master broad skills. The expectation here is a type of social resource. Based on the rules of reciprocity and equivalence [30], employees are obliged to make contributions to their organizations according to what they have received.

Via extensive selection, the selected nurses would experience that organizations value their personal characteristics, abilities and knowledge, and experiences. Participative decision-making and high compensation reflect the importance of the employees to organizations [32]. Thus, they could feel enhanced self-importance, self-image, and confidence in front of difficulties [28]. Career development and employee ownership programs also deliver the sense that a firm depends on its employees for its development. When implementing such practices, an organization’s trust is likely to be perceived by nurses [36].

Practices such as results-oriented performance appraisal and employee participative decision-making do not limit employees’ work to certain posts. Employees enjoy relatively high autonomy under the management of these practices [33]. Extensive training represents the organization’s expectation of enhancing and broadening a nurse’s ability and skill development [35]. Team-oriented appraisal and profit-sharing programs cultivate a sense of shared responsibilities among co-workers. Although team-oriented appraisal might give birth to problems such as free-riding. HPWS as a system of HRM practices is able to seize such detrimental effects. First, the result-oriented appraisal extends requirements and recognition of higher performance. Second, practices such as extensive training mean the investment of the organization in employees. Employees will feel expected to generate better work performance in return. Third, when experiencing profit-sharing practices, employees will perceive that the organization treats them as owners. Therefore, the free-riding effects can be substantially reduced. To sum up, nurses can perceive the self-image, trust, empowerment, and shared responsibilities among co-workers. They also tend to eliminate the detrimental effects, such as free-riding. In reciprocity, they would demonstrate proactivity, great energy, and confidence in problem-solving [36,37]. Therefore, when experiencing HPWS, nurses are likely to be able to positively adapt to traumatic or adverse experiences, i.e., resilience at work [6]. Thus, we propose that: 

**Hypothesis** **1:**
*HPWS is positively related to a nurse’s resilience at work.*


### 2.2. The Role of Resilience at Work

From a socially embedded perspective, thriving [38] can be triggered by three types of agentic behaviors that are the engine of thriving: *task focus*, including focusing behaviors and attention on job tasks and responsibilities; *exploration*, representing experimentation, innovation, risk-taking, and discovery to stretch and grow in new directions; and *heedful relating*, which means looking out for one another to heedfully connect to the social/relational environment [39,40]. Resilient nurses at work can generate all these three types of agentic behaviors. First, resilient nurses are likely to be responsible and focused. It was reported that resilience is strongly associated with work engagement [38] and conscientiousness [41]. The deep reason could be that resilience nurses not only cherish the purpose and value of the work, but also know how to deploy their personal strength at work [38]. Second, resilient nurses are likely to conduct exploration behaviors because they have “the capacity to reframe setbacks, maintain a solution focus”, and they hold “onto personal values, deploying personal strengths”. For example, a study found that resilient nurses were free thinkers and had high levels of surgency [41]. Last, resilience nurses can manage to develop heedful relations. Resilient nurses keep a “work style that includes seeking feedback, advice, and support as well as providing support to others”, and they build up and maintain “a pattern of developing and maintaining personal support networks” [40]. Research revealed that resilient nurses had a low level of neuroticism [41]. Such a personality trait enables them to maintain good relationships with co-workers. Therefore, we propose that: 

**Hypothesis** **2a:**
*A nurse’s resilience at work mediates the relationship between the experienced HPWS and thriving at work.*


Nurses with a high level of job burnout easily feel exhausted, show cynicism and feel incompetent at work [41,42]. It has been argued that developing resilience at work is an effective strategy to cope with job burnout [3]. Resilient nurses are likely to have a low level of job burnout for several reasons. First, resilient nurses are less likely to feel exhausted because they are good at “using work and life routines that help manage everyday stressors, maintain work life balance, and ensure time for relaxation”, hence they experience less emotional exhaustion. They also work on “maintaining a good level of physical fitness and a healthy diet” to keep a good physical state. Second, resilient nurses show less cynicism. The internal psychological characteristics of resilience have been further described as including other established constructs, such as self-efficacy, humor, patience, optimism, and/or faith [43]. Such a type of nurse is hardly cynical. Third, resilient nurses are likely to feel competent at work because they are good at deploying personal strength, reframing setbacks, and maintaining a solution focus [41]. Therefore, we propose that: 

**Hypothesis** **2b:**
*A nurse’s resilience at work mediates the relationship between the experienced HPWS and job burnout.*


Overall, the purpose of this study examines whether HPWS can impact a nurse’s thriving at work and job burnout through resilience at work. The conceptual model of this research is presented in Figure 1.

## 3. Materials and Methods

### 3.1. Procedure and Participants

First, to ask for consent to access hospitals for data collection, an invitation letter and the sample questionnaires were sent to the HRM heads of public hospitals. An agreement was reached with 20 hospitals. Three questionnaires were used to collect data. One thousand and six hundred nurses from the twenty hospitals were invited to participate in this research via the first online questionnaire in May 2019 (the first round of data collection). This first questionnaire contains questions about demographic information and a nurse’s experiences with HPWS. One thousand one hundred and twenty-seven nurses completed the questionnaire. The response rate was 70.44%. The second round of data collection via the second online questionnaire was conducted 3 weeks later, and 987 completed questionnaires were returned. The response rate was 61.7%. The second questionnaire contains questions about a nurse’s resilience at work. Three weeks later, the third round of data collection was implemented. Eight hundred and forty-five nurses completed the third online questionnaire. The response rate was 52.8%. The third questionnaire contained questions about a nurse’s thriving at work and job burnout. Before we conducted the first round of data collection via an online questionnaire, the HR department of hospitals assigned every nurse a unique 3 digit ID code to identify the hospital, sector, and nurse. In total, the research received 845 paired responses. The final response rate was 52.8%.

### 3.2. Measurements

The scales used in the questionnaires are mature scales developed in the literature. A 7-point Likert scale (1 = strongly disagree, 7 = strongly agree) was applied to all the items in the scales. High-performance work systems (HPWS) were measured with Takeuchi and colleagues’ [37] scale containing 21 items. Employees may perceive or experience differences in exposure to work practices [44]. The experience-based, rather than observation-based or description-based, measurement of HRM practices has a stronger relationship with employee outcomes [45]. Therefore, we measure HPWS by using employee-experienced HPWS. Nurses were asked to “evaluate their own experiences of high-performance work practices”. A sample item is “I received training continuously.” It is widely accepted in extant research that researchers develop an index of HPWS by calculating the mathematic mean value of all the scores of HPWS scale items [26,46]. This research employed this way and formed an index of HPWS. The Cronbach’s alpha value of HPWS was 0.833. Resilience at work was measured by using the 20-item scale developed by [40]. The Cronbach’s α value of the scale was found to be 0.819. A sample item is “I have developed some reliable ways to relax when I am under pressure at work”. We used Porath and associates’ scale to measure thriving at work [14]. The scale contains 10 items. A sample item is “I see myself continually improving”. The Cronbach’s α value of the scale was 0.850. We used Maslach and colleagues’ 22-item scale to measure job burnout [42]. A sample item is “I feel fatigued when I get up in the morning to face another day at work”. The Cronbach’s α value of the scale was 0.896. We also included several control variables in the data analysis. They were gender, age, education background, tenure, and working sector in the hospital.

### 3.3. Ethics

We obtained ethics approval from the research ethics committee of the business school where the research project was funded. Second, the top management team’s or HRM head’s official consent to access each of the hospitals was obtained before data collection. Third, each nurse could choose to participate or not in the research. Finally, participants were assured that the data would be used for academic research only and no individual information would be released.

### 3.4. Data Analysis

The data were analyzed with the SPSS software package. We first examined the mean values, standard deviations, and bivariate correlations of the variables in the study (see Table 1 for details).

Then, we tested the hypothesized relationships with hierarchical multiple regressions. In the first step, the mediator, a nurse’s resilience at work, was regressed on control variables and HPWS. The results were shown in model 1 in Table 2. In the second step, a nurse’s thriving at work and job burnout were regressed on the control variables and HPWS (see models 2 and 4 in Table 2 for details). In the last step, a nurse’s thriving at work and job burnout were regressed on control variables and the mediator, thriving at work, (see models 3 and 5 in Table 2 for details).

## 4. Results

Table 1 demonstrates descriptive analysis results, including the mean value, standard deviation, and bi-variate correlation coefficient values of the variables employed in the analysis. All the variables were centered to reduce non-essential multicollinearity [47,48]. The regression analysis results can be seen in Table 2. Multicollinearity is not a concern, since the smallest VIF is 1.007, while the largest is 3.838.

Hypothesis 1 states that HPWS is positively related to a nurse’s resilience at work. In Table 2, the result in model 1 showed that HPWS was positively related to a nurse’s resilience at work (β = 0.500, *p* < 0.001). Therefore, hypothesis 1 was supported.

Hypothesis 2a states that a nurse’s resilience at work mediates the relationship between the experienced HPWS and thriving at work. The result in model 2 demonstrated that HPWS is positively related to thriving at work (β = 0.414 < 0.001). The result in model 3 demonstrated that resilience at work is positively related to thriving at work (β = 0.442, *p* < 0.001). When HPWS and resilience at work were entered into the analysis together, it was found that the positive influence of HPWS on thriving at work became weaker but remained statistically significant (β = 0.259, *p* < 0.001, model 4). Resilience at work was still significantly and positively related to thriving at work (β = 0.310, *p* < 0.001, model 4). Therefore, hypothesis 2a was supported.

Hypothesis 2b states that a nurse’s resilience at work mediates the relationship between the experienced HPWS and job burnout. The result in model 5 demonstrated that HPWS is negatively related to job burnout (β = −0.561 < 0.001). The result in model 3 demonstrated that resilience at work is negatively related to job burnout (β = −0.361, *p* < 0.001). When HPWS and resilience at work were entered into the analysis together, it was found that the negative influence of HPWS on job burnout became weaker but remained statistically significant (β = −0.511, *p* < 0.001, model 7). Resilience at work was still significantly and negatively related to job burnout (β = −0.100, *p* < 0.001, model 7). Therefore, hypothesis 2b was supported.

## 5. Discussion

The purpose of this research is to explore and examine whether HPWS can impact a nurse’s thriving at work and job burnout via resilience at work. Our empirical evidence supported the hypotheses of this study and can generate both theoretical and practical implications.

Theoretically, it generates at least two implications. On the one hand, it was found that HPWS is significantly related to a nurse’s resilience at work. This finding is similar to previous studies. Some researchers reported a series of management policies that could enhance resilience at work, such as tailored training programs [8], moderate-to-vigorous physical activity [9], learning organizational culture [24], etc. Yet, these studies are piecemeal based. The current study tested the effects of the systematic use of HRM practices on resilience at work. The finding of the research is in line with the argument of strategic HRM researchers that HRM practices are more effective when used as a system [49]. A study found that HPWS is positively related to resilience at work in China’s bank industry from a JD-R (job demand-requirement) model perspective [34]. A similar investigation reported a positive relationship between HPWS and resilience at work in the service sector [47]. The empirical evidence of this study supports the social exchange principles. That HPWS as a management tool can intervene in the social exchanges between organizations and nurses, by offering higher-order social resources to induce a nurse’s higher-order pursuit of developing resilience at work. Therefore, it also enriches our understanding of the effects of HRM systems on resilience at work by offering another theoretical explanation of the effects.

On the other hand, it was found that resilience at work partially mediated the influences of HPWS on thriving at work and job burnout, yet, in different directions. In a nurse-focused study, although the literature has reported that resilience at work can enhance positive employee outcomes [22,23,24] and reduce negative employee outcomes [21,23], there has been no research to explore and test whether HPWS can impact employee outcomes through developing resilience at work. Our research filled this gap. Therefore, we expand the validity of employee resilience between HRM systems and employee outcomes by applying it in the nurse management domain. More importantly, our finding stresses the significance of resilience at work as a key individual-level variable to enhance positive employee outcomes, namely, thriving at work, and reduce negative employee outcomes, namely, job burnout simultaneously. Negative and positive employee outcomes are likely to coexist. Our model and findings implicate that developing nurses’ resilience at work is an effective coping strategy that can impact different employees in different directions.

Our finding can also generate practical implications for healthcare organizations in managing nurses. First, a nurse’s resilience at work is crucial in obtaining employee outcomes that are likely to benefit the organization and nurse performance, while reducing outcomes that are likely to harm the organization and nurse performance. Healthcare organizations are suggested to recruit and select resilient candidates and it is also pivotal for both healthcare organizations and schools to cultivate the resilience of nurses and students in nursing. Second, apart from tailored training programs, HPWS could be a useful management tool to cultivate a nurse’s resilience at work. Furthermore, the utilization of HPWS can be further stressed. HPWS was also found to be positively related to thriving and negatively linked to job burnout directly, despite the existence of partial mediation effects of resilience at work.

## 6. Conclusions

The conceptualization and empirical results may enhance the understanding of the crucial relationships between HPWS, resilience at work, thriving at work, and job burnout among nurses in Chinese public hospitals. On the one hand, resilience at work is a type of individual ability that can enhance a nurse’s thriving at work while reducing job burnout. On the other hand, HPWS as a popular type of HRM system used in managing nurses in the researched hospitals can help develop nursing resilience at work, which enhances nurses thriving at work and reduce job burnout.

## 7. Limitations and Suggestions for Future Research

Although this research can generate both theoretical and practical implications, it also has several limitations to acknowledge. Firstly, although the study used time-lagged, it employed a single-respondent approach. Future research is urged to use multiple sources when collecting data. For example, HRM systems and employee outcomes could be evaluated by nurses, coworkers and leaders together. Secondly, the study found that HPWS can enhance thriving at work and reduce job burnout through resilience at work via impacting a nurse’s resilience at work. Other employee outcomes, especially some objectively evaluated outcomes, for instance, patient satisfaction rate, attendance rate, and health conditions, could be employed in future research to further test the role of a nurse’s resilience at work. Lastly, this study’s evidence supports the mediation role of a nurse’s work resilience. Future studies are suggested to explore and examine the boundary conditions of this mediation mechanism. The boundary conditions could include both organizational variables, such as ownership, strategy, and leadership, and individual characteristics, such as gender, tenure, personality, etc. By so doing, research may be able to offer more precise suggestions about when to use HPWS to cultivate resilience at work to further impact employee outcomes.

## Figures and Tables

**Figure 1 healthcare-10-01935-f001:**
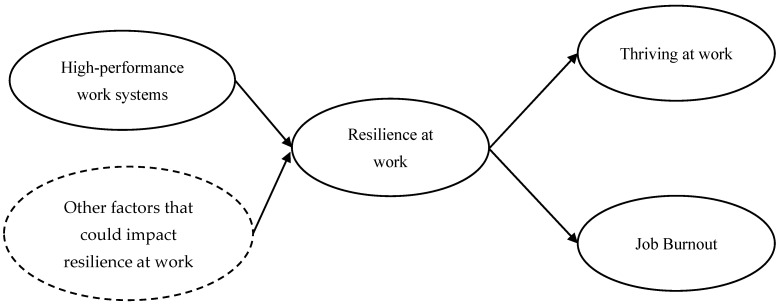
The conceptual model of the mediation role of resilience at work between HPWS and striving at work and job burnout among nurses (note: other factors reflect various factors that can impact resilience at work, such as work-life balance, person-job fit, and affective and emotional states, etc. This research lists “other factors that could impact resilience” in the conceptual model to recognize the potential influences they might exert. However, the focus of this research is to examine the influence of high-performance work systems on resilience at work and thriving at work and job burnout. When testing the hypotheses, the research holds the assumption that everything else is idealistically kept constant).

**Table 1 healthcare-10-01935-t001:** Means, standard deviations and correlations between variables.

	Mean	S.D.	1	2	3	4	5	6	7	8
Nurse’s age	40.48	9.59								
Job tenure	19.89	11.29	0.828 **							
Nurse’s education background	---	---	0.099 **	−0.143 **						
Nurse’s gender	---	---	0.081 *	0.074 *	0.161 **					
Hospital Sector	---	---	−0.039	−0.010	0.021	0.042				
HPWS	5.23	0.80	0.150 **	0.105 **	0.037	0.062	0.029			
Resilience at work	5.60	0.83	0.030	−0.015	−0.039	0.110 **	−0.012	0.136 **		
Thriving at work	4.77	0.86	−0.054	−0.103 **	0.006	0.059	0.023	0.402 **		
Job burnout	3.16	0.82	0.086 *	0.082 *	0.125 **	0.160 **	0.038	−0.525 **	−0.157 **	−0.277 **

*Note: N* = 845; SD = standard deviation. * *p* < 0.05 (2 tailed).; ** *p* < 0.01 (2 tailed).

**Table 2 healthcare-10-01935-t002:** The mediation effects of resilience at work between HPWS and thriving at work and job burnout.

	Resilience at Work	Thriving at Work			Job Burnout		
	M1 Standardized β	M2 Standardized β	M3 Standardized β	M4 Standardized β	M5 Standardized β	M6 Standardized β	M7 Standardized β
**Control Variable**							
Nurse’s age	0.014	0.049	0.082	0.045	0.080	0.008	0.082
Job tenure	−0.033	−0.198 ***	−0.196 ***	−0.187 ***	0.080	0.093	0.077
Nurse’s education background	−0.067 *	−0.051	−0.022	−0.030	0.123 ***	0.099 *	0.116 ***
Nurse’s gender	0.182 ***	0.052	−0.019	−0.005	0.160 ***	0.207 ***	0.178 ***
Hospital Sector	−0.016	0.010	0.023	0.015	0.049	0.030	0.047
**Main Effect**							
HPWS	0.500 ***	0.414 ***		0.259 ***	−0.561 ***		−0.511 ***
**Mediators**							
Resilience at work			0.442***	0.310 ***		−0.361 ***	−0.100 **
**F value**	57.062 ***	32.108 ***	36.383***	40.954 ***	75.479 ***	28.377 ***	66.670 ***
**R^2^**	0.285 ***	0.187 ***	0.201***	0.249 ***	0.346 ***	0.163 ***	0.353 ***
***△*R^2^**	0.243 ***	0.166 ***	0.186***	0.068 ***	0.306 ***	0.124 ***	0.007 **

*Note**: N* = 845, * *p* < 0.05, ** *p* < 0.01, *** *p* < 0.001; VIF value ranges from 1.007 to 3.838.

## Data Availability

The data used to support the findings of this study are available from the corresponding author upon reasonable request.

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
