# Peer review of "High-Performance Work Systems, Thriving at Work, and Job Burnout among Nurses in Chinese Public Hospitals: The Role of Resilience at Work"

_healthcare, 2022, doi:10.3390/healthcare10101935_

Round 1
Reviewer 1 Report
General Points:
In line 48-49, the authors state that "no research has ever explored and examined the impact of the HRM systems on nurse's resilience at work". This is their research gap that they try to fill. I am not entirely sure whether this statement is really true - may be for English-written academic journals (However checking Scholar.google appears to show that in the general field studies of HRM with respect to nurses have been made). I am not sure at all, whether such a statement is true when checking the academic literature in other languages, e.g. German or Spanish.
Lines 102 to 103: Does this take into account problems of potential free-riding of team members at the expense of the other team members - at least potential problems should be mentioned.
Lines 135 to 136: Not all potential problems of nurses can be as easily resolved as suggested in this paragraph. Missing resilience of nurses may stem also from their personal problems in their families/at home with repercussions to their behaviour at the workplace (e.g. husbands that treat them badly, problems with children etc.). Entirely neglecting such aspects instead of at least mentioning such factors is not a very good idea for sound academic analysis.
In other words, one could/should add also in figure 1 in lines 151-152 a further arrow that include "other factors" which can affect resilience at work. At least one should mention the assumption that "everything else" is idealistically kept constant (ceteris paribus). Taking this into account, can lead to a more realistic final assessment of the results found.
However, the cautious conclusions in lines 279 to 283 are fine (..."can enhance".... "can help"..).
Further minor points:
Careful literature review and large academic literature list, however: missing uniformity in quatation of journals, sometimes with, sometimes without dots at abbreviations.
Line 16: Replace "wisdoms" perhaps better with "knowledge".
Line 76: "return" instead of "returns" - this in one example of a few similar small language mistakes; in particular on should check how to write "nurses' " or "a nurse's". Such minor spelling mistakes should be checked by the authors themselves again before handing in the manuscript finally.
Line 117 - I did not find "agentic" in my dictionary, only "agent".
Reviewer 2 Report
Abstract
The abstract has been well written; however, there is a need to remove the sub-headings such as “ (1) Background”, “ (2) Method:” and others.
Chapter 1- Introduction
The introduction chapter has been well written. However, the author(s) should include a problem statement on why there is a need to study job burnout and thriving at work among nurses in China.
Chapter 2 – Literature Review
This section has also been well-written.
Chapter 3- Methodology
This section has also been well-written. However, the author(s) should further explain how the time lagged survey was carried out. How did you ensure that the same set of respondents answered the questions each time?
Why is it necessary to have 2 dependent variables?
Chapter 4- Results.
Table 2 title might need some amendments.
Chapter 5- Discussion and Conclusion
In the discussion section, please expound on the theoretical and practical implications.
The future direction can be further elaborated as some statements are ambiguous.
